# On the Affinities and Systematic Position of *Lachnaeus* Schoenherr and *Rhinocyllus* Germar in the Tribe Lixini (Coleoptera: Curculionidae: Lixinae) Based on the Morphological Characters of the Immature Stages

**DOI:** 10.3390/insects12060489

**Published:** 2021-05-24

**Authors:** Jiří Skuhrovec, Roberto Caldara, Rafał Gosik, Filip Trnka, Robert Stejskal

**Affiliations:** 1Group Function of Invertebrate and Plant Biodiversity in Agro-Ecosystems, Crop Research Institute, Drnovská 507, 6 Ruzyně, CZ-161 06 Praha, Czech Republic; 2Center of Alpine Entomology, University of Milan, Via Celoria 2, 20133 Milan, Italy; roberto.caldara@gmail.com; 3Department of Zoology and Nature Protection, Institute of Biological Sciences, Maria Curie-Skłodowska University, Akademicka 19, 20-033 Lublin, Poland; r.gosik@poczta.umcs.lublin.pl; 4Department of Ecology & Environmental Sciences, Faculty of Science, Palacký University Olomouc, Šlechtitelů 27, CZ-783 71 Olomouc, Czech Republic; filip.trnka88@gmail.com; 5Administration of Podyji National Park, Na Vyhlídce 5, CZ-669 02 Znojmo, Czech Republic; stejskal@nppodyji.cz

**Keywords:** Coleoptera, Curculionidae, Lixini, *Lachnaeus*, *Rhinocyllus*, morphology, larva, pupa, description

## Abstract

**Simple Summary:**

The detailed descriptions of immature stages of *Lachnaeus crinitus* Schoenherr, 1826, *Rhinocyllus alpinus* Gültekin, Diotti and Caldara, 2019 and *R. conicus* (Frölich, 1792), belonging to the Lixini (Curculionidae: Lixinae), are very important for an increased understanding of the relationship between the species and of the taxonomic and phylogenetic value of the tribes and genera in the Lixinae. The complex of these new data has allowed us to support that (1) *Lachnaeus* and *Rhinocyllus* are two valid genera that are different from *Larinus*, (2) Rhinocyllini is not a tribe different from Lixini, and (3) the separation of *Rhinocyllus* into two subgenera is rational. In the tribe Lixini, these new data may have an important role, because *Rhinocyllus conicus* and several other species of the genera *Lixus* and *Larinus* have a practical or at least potential use as biological control agents against invasive and noxious weeds.

**Abstract:**

Mature larvae and pupae of *Lachnaeus crinitus* Schoenherr, 1826 and *Rhinocyllus alpinus* Gültekin, Diotti and Caldara, 2019 and pupae of *R. conicus* (Frölich, 1792), belonging to the Lixini (Curculionidae: Lixinae), are morphologically described for the first time. They possess all the characters considered distinctive in the immature stages of this tribe and are distinguishable from all the related genera by a combination of some characters (e.g., presence of endocarina, shape of premental sclerite; the number of *pds* on the abdominal segments; size and presence of urogomphi). It is emphasized that the controversial tribe Rhinocyllini is not supported by the characters of the larvae and pupae of *Rhinocyllus* and that the two subgenera of this genus, *Rhinocyllus* s. str. and *Rhinolarinus*, are separable from each other not only by characters of the adult but also distinctive characters of the larvae and pupae. These results confirm that the morphology of the immature stages, which is usually overlooked, can be very important for the purpose of identifying new characters that are useful for clarifying taxonomical and phylogenetic complex situations based only on the study of the imagoes.

## 1. Introduction

The cosmopolitan subfamily Lixinae of Curculionidae includes approximately 90 genera and 1500 species [1,2], with the largest number of taxa present in the Palaearctic and Afrotropical regions. It is important to note that several species in this subfamily belonging to well-known genera such as *Lixus* Fabricius, 1801 (e.g., *L. cardui* Olivier, 1807; *L. filiformis* (Fabricius, 1781) and *L. cribricollis* Boheman, 1836), *Larinus* Germar, 1824 (e.g., *L. minutus* Gyllenhal, 1835) and *Rhinocyllus* Germar, 1817 (e.g., *R. conicus* (Frölich, 1792)) have a certain relevance as potential biological control agents against invasive Asteraceae of the genera *Carduus*, *Cirsium*, *Centaurea*, *Onopordum* and *Tanacetum* [3,4,5]. There is no complete agreement in the division of this subfamily in tribes. Whereas all the authors are concordant in accepting two tribes, Lixini and Cleonini, a third tribe, Rhinocyllini, is controversial. This tribe was proposed by Lacordaire [6] for the genera *Rhinocyllus* and *Microlarinus* Hochhuth, 1847 and briefly described with the following sentence: “Rostrum at the most as long as the head, angulate and dorsally flat; scrobes anteriorly complete”. His opinion was followed by Capiomont [7], Petri [8] and Csiki [1], who included the genus *Bangasternus* Des Gozis, 1886 in this tribe and newly placed *Microlarinus* into Lixini. More recently, the tribe Rhinocyllini, including all three genera, was accepted by Pesarini [9], Alonso-Zarazaga and Lyal [2], Meregalli [10], and Arzanov and Grebennikov [11]. Conversely, *Rhinocyllus* and *Bangasternus* were assembled in Lixini by Reitter [12,13], Hustache [14], Hoffmann [15], Ter-Minasian [16], Colonnelli [17], Gültekin [18] and Gültekin and Fremuth [19], who did not find clear morphological or ecological boundaries between Lixini and Rhinocyllini, especially between small species of *Larinus* of the subgenus *Larinomesius* Reitter, 1924 and Rhinocyllini.

The recent description of *Rhinocyllus alpinus* Gültekin, Diotti and Caldara, 2019 and its placement in the new subgenus *Rhinolarinus* Gültekin, Diotti and Caldara, 2019 strengthened the latter opinion and set very interesting taxonomic questions. The adults of this species lack some characters previously considered important in the definition of the genus *Rhinocyllus* and its separation from *Larinus*, making the establishment of the correct taxonomic position of this new species questionable. This regards the presence of deep lateral sulci and distinct ridges on the dorsum of the rostrum laterally, which are completely missing in *R. alpinus*. However, the rostrum at its sides is still angulated and not beveled, whereas a short sulcus is visible on the head around the upper margin of the eyes. The differences between this species and *Larinus*, e.g., *L. obtusus* Gyllenhal, 1835, are surely less evident than for the other species of *Rhinocyllus* but are still consistent; a character useful for separating these two genera remains the shape of tergite VIII in the female. This structure was investigated by Gültekin [18], who pointed out its significant value for classification, as it is different in shape in *Rhinocyllus*, *Bangasternus* and *Larinus*.

Another genus included in the Lixini, but not subject to careful study on its relationships with other genera, is *Lachnaeus* Schoenherr, 1816, which is composed of only four species with a central Asian/Mediterranean distribution. The adults seem more related to *Larinus* s.l., from which they primarily differ by the very long setae covering the dorsal integument and the legs.

It is well known that the study of the morphology of the immature stages can be of much help in the generic taxonomic relationships within weevil subfamilies and tribes; this seems true also for Lixinae, as recent descriptions of several species of the main genera *Lixus* and *Larinus* have shown [20,21,22].

In this regard, we recently had the opportunity to study the immature stages of *Rhinocyllus conicus*, *R. alpinus*, and *Lachnaeus crinitus* Schoenherr, 1826. Therefore, the aims of this paper were (1) to describe pupae of *R. conicus* and larvae and pupae of the other two species for the first time, (2) to report the differences and affinities of the immature stages of these species with regard to the other genera of the subfamily, (3) to establish whether the tribe Rhinocyllini is taxonomically distinct from Lixini, and (4) to establish whether the subgenus *Rhinolarinus* is also distinguishable from *Rhinocyllus* s.str. by the characters of the immature stages.

## 2. Materials and Methods

### 2.1. Material Sampling

All specimens used in this study were collected inside in the inflorescences of the host plants contemporaneously to their imagoes feeding on flowers or in copula. Some of the larval and pupal materials were preserved in Pampel fixation liquid [23] and used for morphological descriptions. The remaining specimens were deposited in the collection of Group Function of Invertebrate and Plant Biodiversity in Agro-Ecosystems of the Crop Research Institute (Prague, Czech Republic). The insect-host plant were identified by the same collectors of the weevil specimens. 

### 2.2. Morphological Description

All described specimens were examined under an optical stereomicroscope (Olympus SZ 60 and SZ11 (Olympus Inc. Tokyo, Japan)), with calibrated oculars. The following characteristics were measured for each larva: width of the head (HW), length of the body (BL; larvae fixed in a C-shape were measured in the middle of the segments in lateral view), and width of the body at the widest place (BW, i.e., at the first abdominal segment). For the pupae, the length (BL), thorax width (THW), and width of the body were determined at the widest place (BW; at the level of the middle legs). The results of the measurements of all specimens are shown in Table 1. Drawings and outlines were made using a drawing tube (MNR–1) installed on a stereomicroscope (Ampliwal, (Amplival Pol-d, Carl Zeiss Jena, Germany)) and were processed using computer software (Corel Photo–Paint X7, Corel Draw X7, (Corel Inc. Austin, TX, USA)). The numbers of setae of bilateral structures are given for one side.

Slide preparation followed by May [24]. The larvae selected for study were cleared in 10% potassium hydroxide (KOH), rinsed in distilled water and dissected under the microscope. After clearing, the head, mouthparts, and body (thoracic and abdominal segments) were separated and mounted on permanent microscope slides in Faure–Berlese fluid (50 g of gum arabic and 45 g of chloral hydrate dissolved in 80 g of dissolved water and 60 cm^3^ of glycerol) [25].

Photos were taken using an Olympus BX63 microscope and processed in Olympus cellSens Dimension software (Olympus Inc. Tokyo, Japan). The larvae selected for pictures using SEM (scanning electron microscope) were first dried in absolute ethyl alcohol (99.8%), rinsed in acetone, treated by CPD (critical point drying), and then gold-plated. A TESCAN Vega 3 SEM (Tescan, Brno, Czech Republic) was used for the examination of selected structures.

The terms and abbreviations used here for the setae of the mature larvae and pupae were obtained from Anderson [26], May [24], Marvaldi [27,28] and Skuhrovec et al. [29], with antennae terminology following Zaharuk [30].

All morphological abbreviations used in the text are as follows:

**Abd. I–X**–abdominal segments 1–10, **Th. I–III**–thoracic segments 1–3, **at**–antenna, **clss**–clypeal sensorium, **ds**–digitiform sensillum, **st**–stemmata, **se**–sensorium, **sa**–sensillum ampullaceum, **sb**–sensillum basiconicum, **snp** sensillae pore, **ss**–sensillum styloconicae, tra–terminal receptive area, **lr**–labral rods, **ur**–urogomphus; setae: ***als***–anterolateral, ***ams***–anteromedial, ***as***–alar (larva), ***as***–apical (pupa), ***cls***–clypeal, ***d***–dorsal (pupal abdomen), ***des***–dorsal (larval head), ***dms***–dorsal malar, ***ds***–discal (pupal prothorax), ***ds***–dorsal (larval abdomen), ***eps***–epipleural, ***es***–epistomal, ***eus***–eusternal, ***fs***–frontal, ***les***–lateral epicranial, ***ligs***–ligular, ***lrs***–labral, ***ls***–lateral, ***lsts***–laterosternal, ***mbs***–malar basiventral, ***mds***–mandibular, ***mes***–median, ***mps***–maxillary palp, ***pda***–pedal, ***pds***–postdorsal, ***pls***–posterolateral, ***pes***–postepicranial, ***pfs***–palpiferal, ***pms***–postlabial, ***prms***–prelabial, ***prns***–pronotal, ***prs***–prodorsal, ***ps***–pleural, ***sls***–super lateral, ***sos***–superorbital, ***ss***–spiracular, ***stps***–stipal, ***sts***–sternal, ***ves***–ventral, ***vms***–ventral malar, **vs**–vertical.

## 3. Results

### 3.1. The Morphology of Immature Stages of Lachnaeus crinitus

#### 3.1.1. Material Examined 

Larvae: Czech Republic, Pohansko, *Inula britannica*, 2 exx., 14.08.2014, leg. Filip Trnka; Hnanice, *Inula britannica*, 3 exx., 6.08.2014, leg. Robert Stejskal.

Pupae: Czech Republic, Pohansko, *Inula britannica*, 2 ♂ exx, 14.08.2014, leg. Filip Trnka; Hnanice, *Inula britannica*, 1 ♀ ex., 3 ♂ exx, 06.08.2014, leg. Robert Stejskal.

**Figure 1 insects-12-00489-f001:**
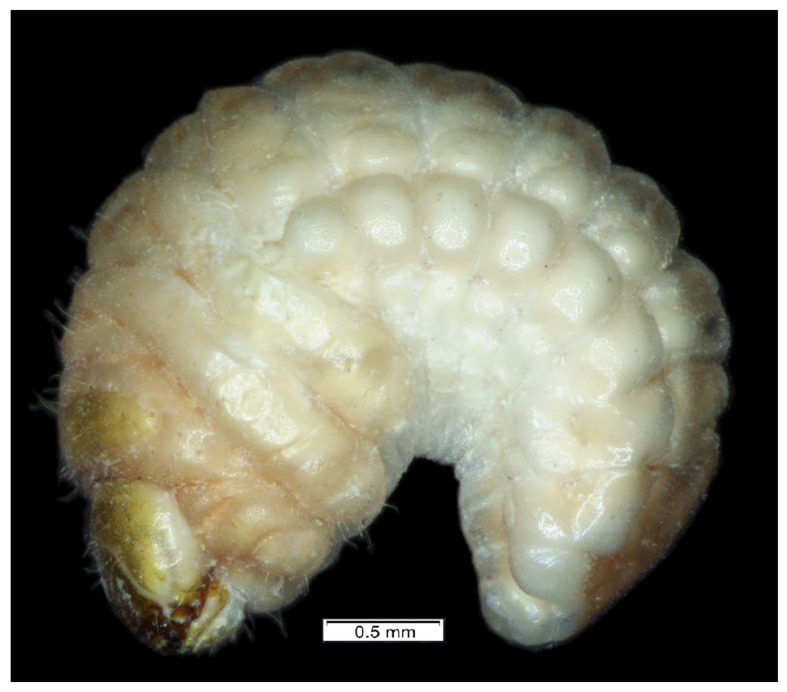
*Lachnaeus crinitus* habitus of mature larva, lateral view.

#### 3.1.2. Description of Mature Larva

*Measurements* (in mm): body length: 2.50 to 4.25. Body width (mid abdomen): up to 1.20. Head width: 0.90 to 1.75. All results of measurements are given in Table 1.

*General features*: body (Figure 1) stout, C-shaped, rounded in cross-section. Cuticle of the body densely covered with asperities (Figure 1).

*Coloration.* Dorsal parts of all pronotal and abdominal segments I–IX yellowish pigmented, rest of the body greyish (Figure 1). Head light yellow (Figure 1), frons dark yellow pigmented. 

*Vestiture.* Setae on body of various lengths from medium to minute, but all of hair form.

**Figure 2 insects-12-00489-f002:**
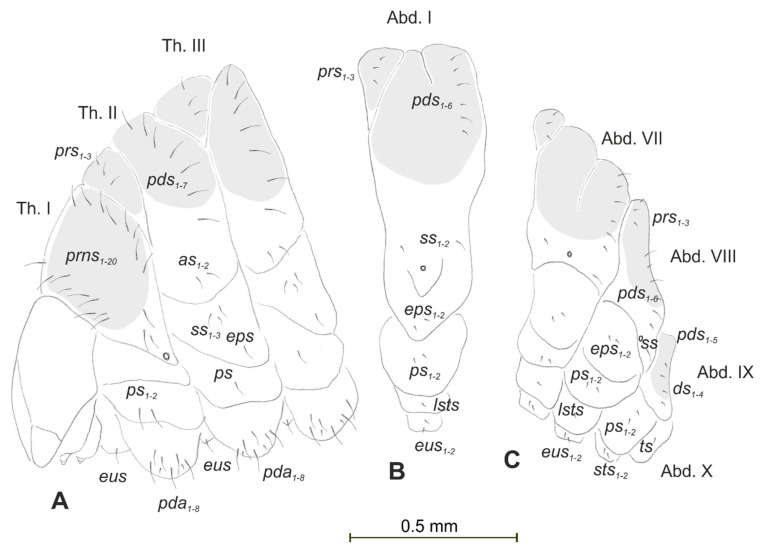
*Lachnaeus crinitus* mature larva, chaetotaxy. (**A**)—lateral view of thorax; (**B**)—lateral view of abdominal segment I; (**C**)—lateral view of abdominal segments VII–X (Th. I–III—thoracic segments 1–3, Abd. I–X—abdominal segments 1–10, setae: *as*—alar, *ds*—dorsal, *ps*—pleural, *eps*—epipleural, *eus*—eusternal, *pda*—pedal, *pds*—postdorsal, *prns*—pronotal, *prs*—prodorsal, *ss*—spiracular, *sts*—sternal, *ts*—terminal).

Body prothorax prominent. Meso- and metathorax almost equal in size, each divided dorsally into two lobes (prodorsal lobes medium, postdorsal lobes prominent). Pedal lobes of thoracic segments conical (Figure 2A). Abdominal segments I–II of similar size, the largest segments IIII–IX tapering towards the posterior body end (Figure 2B,C). Abdominal segments I–VII, each with a relatively narrow prodorsal lobe and always well developed, divided into two equal postdorsal lobes. Dorsal part of abdominal segment VIII undivided. Epipleural, pleural, and laterosternal lobes of segments I–VIII conical,
well-isolated. Abdominal segment IX divided into large dorsal, large pleural, and relatively small sternal lobes (Figure 2C). Abdominal segment X divided into four lobes of various sizes (dorsal the largest, ventral the smallest, and lateral lobes equal in size). Anus situated ventrally (Figure 2C). The spiracles of thorax look vestigial and on high magnification appear bicameral (Figure 3A), placed latero-ventrally on prothorax; abdominal spiracles on abdominal segments I–VIII vestigial (Figure 3B), placed medio-laterally. Chaetotaxy of thoracic and abdominal segments well-developed. Prothorax (Figure 2A) with 20 medium *prns*, almost equal in length (17 placed on pronotal sclerite, next three above the spiracle); two medium *ps* and one medium *eus*. Meso- and metathorax each with three short *prs*; seven medium *pds*, equal in length; alar area with two medium *as*; three *ss* of various lengths (two short and one minute); one medium *eps*; one medium *ps* and one medium *eus*. Pedal areas of thoracic segments each with eight *pda* of various lengths. Abdomen (Figure 2B,C): setae on abdomen seem generally shorter than those on thorax. Abdominal segments I–VIII with three minute *prs*; six *pds* (*pds_1_*, *pds_3_* and *pds_5_* medium, *pds_2_*, *pds_4_* and *pds_6_* minute); two *ss* (*ss_1_* minute, *ss_2_* medium; segment VIII with only medium *ss_1_*); two minute *eps*; two minute *ps*; one minute *lsts* and two minute *eus*. Abdominal segment IX with four minute *ds*; two minute *ps* and two minute *sts*. Abdominal segment X with one minute *ts* on each of the lobes (Figure 2C).

*Head capsule* rounded (Figure 4A,B). Endocarinal line absent. Frontal sutures distinct in entire length up to antennae. A single pair of stemma (st) in the form of prominent dark pigmented spots with convex corneas, placed anterolaterally (Figure 4A). Hypopharyngeal bracon without median sclerome. Setae of head of various lengths, elongated to minute. *Des_1_* elongated, placed medially; *des_2_* elongated, placed posterolaterally; *des_3_* elongated, placed above frontal suture; *des_4_* medium, placed anteromedially; *des_5_* elongated placed anterolaterally, and *des_6_* elongated, placed above stemma. *Fs_1_* medium, placed posteriorly; *fs_2_* medium, placed mediolaterally; *fs_3_* medium, placed medially; *fs_4_* elongated, placed anteromedially; and *fs_5_* elongated, placed anterolaterally, close to epistome. *Les_1_* and *les_2_* slightly shorter than *des_1_*. Postepicranial area with four minute *pes* (Figure 4B). Antennae (Figure 5A,B) with oblique positions on each side at anterior margins of the head; membranous basal segment convex, semi-spherical, bearing conical, relatively elongated sensorium and four sensilla ampullaceum (sa). Clypeus (Figure 6A–C) approximately 3.7 × wider than long, *cls_1_* and *cls_2_* medium, equal in length, elongated, sensillum (clss) placed between them; anterior margin of clypeus rounded to inside. Labrum (Figure 6A–C) approximately 1.7 × wider than long, anterior margin strongly sinuated; *lrs_1_* and *lrs_2_* placed anteromedially, *lrs_3_* placed anterolaterally, all *lrs* elongated, equal in length. Epipharynx (Figure 6A,B) with three elongated, finger-like *als*, of various lengths; three finger-like *ams*: *ams_1_* short, *ams_2_* medium, *ams_3_* relatively elongated; *mes_1_* distinctly smaller than *mes_2_* both placed anteriorly; labral rods (lr) kidney-shaped, placed parallel; a single pair of sensillae pores (snp) arranged in the posterior part; surface smooth. Mandibles (Figure 6D) with two apical teeth of unequal height, the inner one subapical and much smaller; an additional protruding protuberance on the cutting edge between the apex and the middle of the mandible; *mds_1_* minute, *mds_2_* medium; both placed laterally in shallow holes. Maxillolabial complex (Figure 7A–D) on stipes with one elongated *stps*, two elongated *pfs* and one minute *mbs*. Mala with a row of six elongated, finger-like *dms* of various lengths and four *vms* (two medium and two short) (Figure 7A–D and Figure 8A). Maxillary palpi with two palpomeres; basal palpomeres, some wider than distal; length ratio of basal and distal palpomeres almost 1:1; basal palpomere with one medium *mps* and two pores, distal palpomere with one pore, one digitiform sensillum (*ds*) and a group of nine apical sensillae (four ampullaceae (*sa*) and five basiconicae (*sb*)) in terminal receptive area (tra) (Figure 8B). Dorsal parts of mala partially covered with fine asperities. Labium with prementum almost triangular shaped, with one elongated *prms* placed medially. Ligula concave, semicircular at margin, with a single minute *ligs.* Labial palpi two-segmented; basal palpomere wider than distal; length ratio of basal and distal palpomeres almost 1:1. Each palpomera with a single pore, distal palpomerae with a group of 14 apical sensillae (five ampullaceae and nine basiconicae) in terminal receptive area (Figure 8C). Premental sclerite U-shaped; postmentum prominent, membranous, triangular, with three *pms* of various lengths: *pms_1_* elongated, located posterolaterally; *pms_2_* elongated, situated mediolaterally and *pms_3_* medium, placed anterolaterally; lateral and posterolateral parts covered with minute asperities.

#### 3.1.3. Description of Pupa

*Measurements* (in mm): body length: 5.00 to 6.50 (mean 6.00). Body width at the widest part: 3.00 to 3.50 (mean 3.40). All results of measurements are given in Table 1.

*Body* stout, slightly curved (Figure 9A–C). Dorsal part dark brown, rest of the body yellowish; cuticle covered with fine asperities, smooth only on head and pronotum (Figure 10D,E). Rostrum short, as long as wide, reaching procoxae. Pronotum twice as wide as long, trapezium-shaped. Mesonotum and metanotum almost equal in size. Abdominal segments I–III of equal length, segments IV–VI tapered gradually towards end of the body, segment VII prominent, segments VIII and IX very small. Spiracles placed dorsolaterally on abdominal segments I–VI; those on segments I–V functional, and that on segment VI vestigial (Figure 10A–C).

*Chaetotaxy* well-visible, with setae of various lengths from medium to short on the dorsal part of the body placed on prominent protuberances, especially on prothorax and abdominal segments VI and VII. Head with one *pas*, two *os* and two *sos*, rostrum without setae. All setae on head equal in length, short (Figure 10A–C). Pronotum with three *as*, three *ls*, two *sls*, three *ds*, and seven *pls* equal in length (Figure 10D). Each, meso- and metathorax with nine minute setae (three placed posteromedially, six along anterior margin). Each femora with two medium, hair-like setae (Figure 10A–C). Abdominal segments I–V with 11 minute setae (three placed posteromedially, eight placed along anterior margin of each of the segments) (Figure 10E). Abdominal segment VI with 11 setae (three minute placed posteromedially, eight short, placed on protuberances along anterior margin of the segment). Abdominal segment VIII with 10 setae: three minute placed posteromedially, four medium placed anteromedially and three short placed anterolaterally, *d_4_*–*d_7_* placed on prominent protuberances (Figure 10F). Abdominal segment VIII with four medium setae, placed anteromedially on small protuberances. 

Each lateral part of abdominal segments I–VIII with two minute setae. Ventral parts of abdominal segments I–VIII without setae. Abdominal segment IX with a pair of short, conical urogomphia (Figure 10G).

### 3.2. The Morphology of Immature Stages of Rhinocyllus alpinus

#### 3.2.1. Material Examined

Larvae (29 exx.) and pupae (16 exx.): Northern Italy, Lombardia (Bergamo province), Valle Brembana San Simone, 1660 m, in the inflorescences of *Cirsium alsophilum* (Pollini) Soldano; 29 June 2018; leg. L. Diotti and R. Caldara.

#### 3.2.2. Description of Mature Larva

*Measurements* (in mm): body length: 3.25 to 5.00 (mean 4.50). Body width (mid-abdomen): up to 3.00. Head width: 1.20 to 1.30 (mean 1.25). All results of measurements are given in Table 1.

*General features*: body stout, C-shaped, rounded in cross-section (Figure 11). Cuticle of the body densely covered with asperities (Figure 11).

*Coloration*: only pronotal sclerites are brownish; the rest of the thoracic and all abdominal segments are white or greyish (Figure 11). Head is brown, covered with irregular dark brown stripes.

*Vestiture*: Setae on body of various lengths from medium to minute, but all of hair form.

**Figure 12 insects-12-00489-f012:**
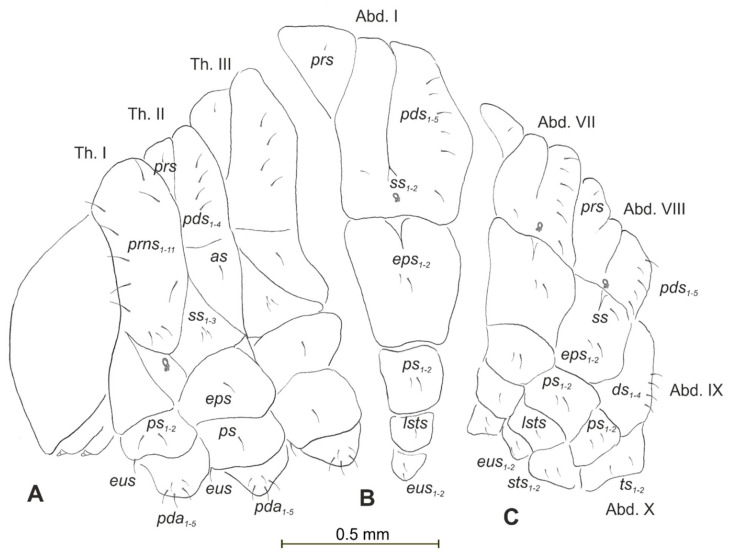
*Rhinocyllus alpinus* mature larva, chaetotaxy. (**A**)—lateral view of thorax; (**B**)—lateral view of abdominal segment I; (**C**)—lateral view of abdominal segments VII–X (Th. I–III—thoracic segments 1–3, Abd. I–X—abdominal segments 1–10, setae: *as*—alar, *ds*—dorsal, *ps*—pleural, *eps*—epipleural, *eus*—eusternal, *pda*—pedal, *pds*—postdorsal, *prns*—pronotal, *prs*—prodorsal, *ss*—spiracular, *sts*—sternal, *ts*—terminal).

Body prothorax is small, with slightly pigmented pronotal shield. Meso- and metathorax are almost equal in size, each divided dorsally into two lobes (prodorsal lobes small, postdorsal lobes prominent). Pedal lobes of thoracic segments are well isolated, conical (Figure 12A). Abdominal segments I–III of similar size, the largest, segments IV–IX tapering towards posterior body end. Abdominal segments I–VIII with a relatively narrow, prodorsal lobe, and always well-developed, divided into two equal-sized parts of postdorsal lobes (Figure 12B,C). Epipleural, pleural and laterosternal lobes of segments I–VIII conical, well-isolated. Abdominal segment IX divided into large dorsal folds and relatively small pleural and sternal lobes (Figure 12C). Abdominal segment X divided into four lobes of various sizes (dorsal the largest, ventral the smallest, and lateral lobes equal in size). Anus situated terminally (Figure 12C). All spiracles bicameral (with two branches); thoracic (Figure 13A) placed latero-ventrally on prothorax; abdominal spiracles (Figure 13B) placed medio-laterally on segments I–VIII. Chaetotaxy of thoracic and abdominal segments well-developed. Prothorax (Figure 12A) with 10 medium and one short *prns* (eight placed on pronotal sclerite, next three above the spiracle); two medium *ps* and one medium *eus*. Meso- and metathorax with one minute *prs*; four medium *pds*, equal in length; alar area with one medium *as*; three *ss* of various lengths (two short and one minute); one medium *eps*; one medium *ps* and one medium *eus*. Pedal areas of thoracic segments with five *pda* of various lengths. Abdominal segments I–VIII (Figure 12B,C) with one minute *prs*; five *pds* (*pds_1_*, *pds_3_* and *pds_5_* medium, *pds_2_* and *pds_4_* short), two *ss* (*ss_1_* minute, *ss_2_* medium; segment VIII with one medium *ss_1_*); two *eps* (one medium and one minute); two *ps* (one medium and one minute); one medium *lsts* and two medium *eus*. Abdominal segment IX with four medium *ds*; two *ps* (one medium and one minute) and two medium *sts*. Abdominal segment X with two *ts* equal in size on each of the lobes (Figure 12C).

*Head capsule* almost rounded (Figure 14A,B). Endocarinal line well-visible, half as long as the frons. Frontal sutures distinct in entire length up to antennae. A single pair of stemma (st) in the form of prominent dark pigmented spots with convex corneas placed anterolaterally (Figure 14A). Hypopharyngeal bracon without median sclerome. Setae of head of various lengths, medium to minute. *Des_1_* placed medially; *des_2_* placed posterolaterally; *des_3_* placed above frontal suture; *des_4_* placed anteromedially; *des_5_* much shorter than the other *des*, placed anterolaterally above the stemma. *Fs_1_* short, placed posteriorly; *fs_2_* short, placed mediolaterally; *fs_3_* short, placed medially; *fs_4_* medium, placed anteromedially, and *fs_5_* medium, placed anterolaterally, close to epistome. *Les_1_* and *les_2_* slightly shorter than *des_1_*. Postepicranial area with five minute *pes* (Figure 14B). Antennae (Figure 15) with oblique positions on each side at anterior margins of head; membranous basal segment convex, semi-spherical, bearing conical, relatively short sensorium and nine sensilla: five basiconica (sb), one styloconica (ss) and three ampullaceum (sa). Clypeus (Figure 16A–C) approximately 2 × wider than long, *cls_1_* short, *cls_2_* elongated, both placed posterolaterally, sensillum (clss) placed close to *cls_2_*; anterior margin of clypeus slightly rounded to inside. Labrum (Figure 16A–C) approximately 2 × wider than long, anterior margin sinuated; *lrs_1_* and *lrs_2_* elongated, both placed anteromedially, *lrs_3_* short, placed laterally. Epipharynx (Figure 16A,B) with three finger-like *als* of various lengths; three finger-like *ams*: *ams_1_* medium, *ams_2_* thin, *ams_3_* short and thick; *mes_1–2_* equal in length, thick; labral rods(lr) elongated, more sclerotized at the apex, almost parallel; a single pair of sensillae pores (snp) arranged in the posterior part; surface smooth. Mandibles (Figure 16D) with two apical teeth of unequal height; single medium *mds* placed laterally in shallow holes. Maxillolabial complex (Figure 17A–D) on stipes with one medium *stps*, two medium *pfs* and one minute *mbs* plus sensillum. Mala with a row of seven finger-like *dms* of various lengths and five *vms* (three medium and two short) (Figure 17A–D and Figure 18A). Maxillary palpi with two palpomeres; basal palpomere much wider than distal; length ratio of basal and distal palpomeres almost 1:1; basal palpomere with one short *mps* and two pores, distal palpomere (Figure 18B) with one pore, one digitiform sensillum (ds) and a group of 12 apical sensillae (ampullaceae) in the terminal receptive area (tra). Dorsal parts of mala partially covered with fine asperities. Labium with a prementum cup shape, with one medium *prms* placed medially. Ligula concave, semicircular at margin, with single minute *ligs.* Labial palpi two-segmented; basal palpomere wider than distal; length ratio of basal and distal palpomeres almost 1:0.7; each palpomerae with a single pore, distal palpomerae with a group of 13 apical sensillae (ampullaceae) in terminal receptive area (Figure 18C). Premental sclerite trident-shaped (median branch well-sclerotized), posterior extension with elongated, blunt apex; postmentum prominent, membranous, triangular, with three *pms* of various lengths: *pms_1_* medium, located posterolaterally, *pms_2_* elongated, situated mediolaterally and *pms_3_* short, placed anterolaterally; lateral and posterolateral parts covered with minute asperities (Figure 18A).

#### 3.2.3. Description of Pupa

*Measurements* (in mm): body length: 5.00 to 6.50 (mean 6.00). Body width at the widest part: 3.00 to 3.50 (mean 3.40). All results of measurements are given in Table 1.

*Body* yellowish, rather stout, cuticle covered with fine asperities, smooth only on head and pronotum (Figure 19A–C). Rostrum 1.2 × longer than wide, reaching procoxae. Pronotum twice as wide as long, sinuated laterally. Mesonotum smaller than metanotum. Abdominal segments I–V of equal length, segments VI–VII tapered gradually towards end of the body, segment VIII semicircular, segment IX very small. Spiracles placed dorsolaterally on abdominal segments I–VI; those on segments I–V functional, and that on segment VI vestigial.

**Figure 19 insects-12-00489-f019:**
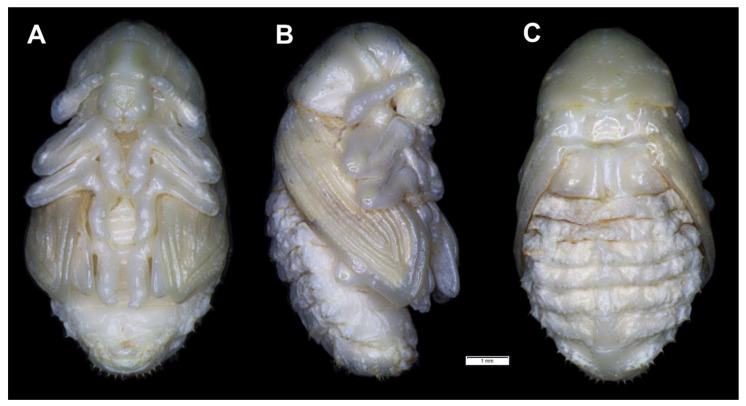
*Rhinocyllus alpinus* habitus of pupa. (**A**)—ventral view; (**B**)—lateral view; (**C**)—dorsal view.

**Figure 20 insects-12-00489-f020:**
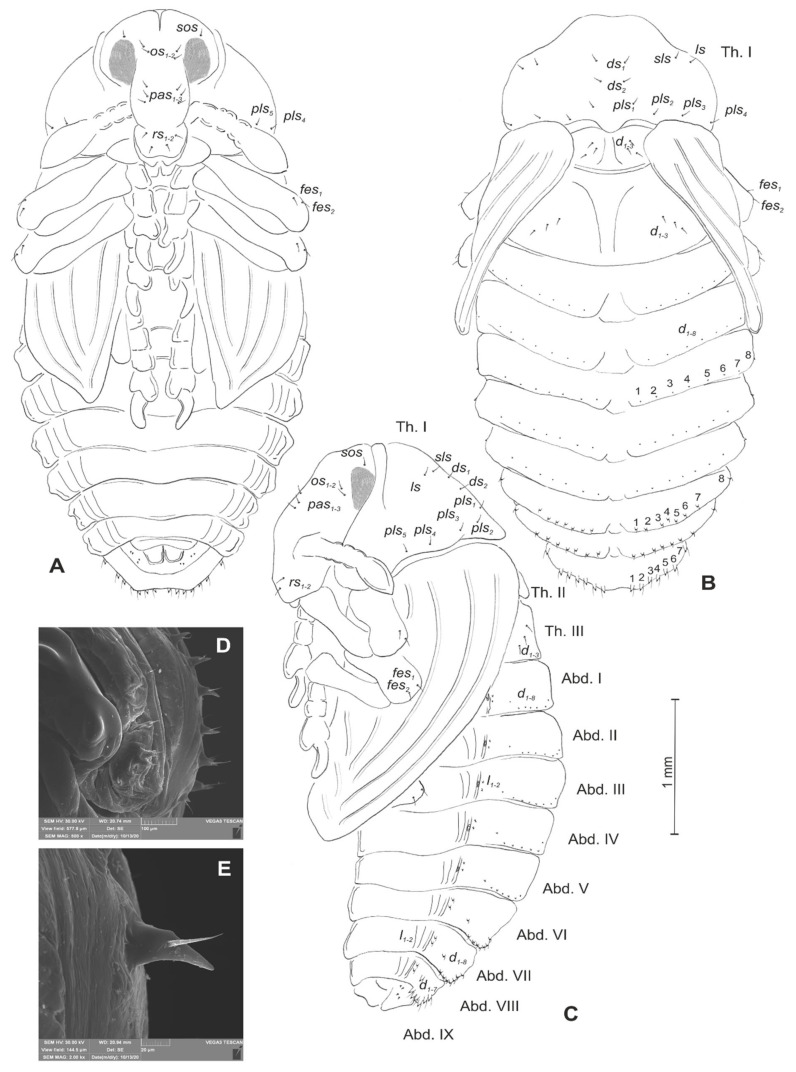
*Rhinocyllus alpinus* pupa. (**A**)—ventral view; (**B**)—dorsal view; (**C**)—lateral view; (**D**,**E**)—Abd. VIII, dorsal view, magnification (Th. I–III—pro–, meso– and metathorax, Abd. I–IX—abdominal segments 1–9, ur—urogomphus, setae: *as*—apical, *d*—dorsal, *ds*—discal, *fes*—femoral, *ls*—lateral, *os*—orbital, *pas*—postantennal, *pls*—posterolateral, *rs*—rostral, *sls*—superlateral).

*Chaetotaxy* visible, with setae of various lengths from medium to short on abdominal segments VI–VIII placed on small (slightly growing on segment VIII) protuberances. Head with three *pas*, two *os* and single *sos*, rostrum with two *rs*. All setae on head and rostrum equal in length and medium (Figure 20A–C). Pronotum with one *ls*, one *sls*, two *ds*, and five *pls* equal in size. Meso- and metathorax with three short setae placed medially on dorsum. Each femora with two medium, hair-like setae (Figure 20A). Abdominal segments I–V with eight minute setae (all placed along anterior margins of each of the segments). Abdominal segments VI and VII with eight short setae placed on thorn-like protuberances. Abdominal segment VIII with seven medium setae placed on thorn-like protuberances (Figure 20D). Abdominal segment IX with four minute setae placed ventrally. Each lateral part of abdominal segments I–VIII with two minute setae. Ventral parts of abdominal segments I–VIII without setae. Urogomphia invisible (Figure 20E).

### 3.3. The Morphology of Immature Stages of Rhinocyllus Conicus

#### 3.3.1. Material Examined

Pupae: Poland, Ciechanki ad Łęczna, *Cirsium rivulare* (Jacq.) All, 5 ♀ exx, 10 ♂ exx., 17 June 2015, leg. R. Gosik.

#### 3.3.2. Description of Pupa

*Measurements* (in mm): body length: 4.50 to 7.00 (mean 5.50). Body width at the widest part: 2.10 to 3.40 (mean 2.90). All results of measurements are given in Table 1. 

*Body* greyish or yellowish, slightly elongated, cuticle covered with fine asperities, smooth only on the head and pronotum (Figure 21A–C). Rostrum almost as wide as long, reaching procoxae. Pronotum 2.25 × wider than long, rounded laterally. Mesonotum and metanotum equal in size. Abdominal segments I–IV of equal length, segments V–VII tapered gradually towards the end of the body, segment VIII semicircular, segment IX small. Spiracles placed dorsolaterally on abdominal segments I–VI; those on segments I–V functional, and that on segment VI vestigial.

**Figure 21 insects-12-00489-f021:**
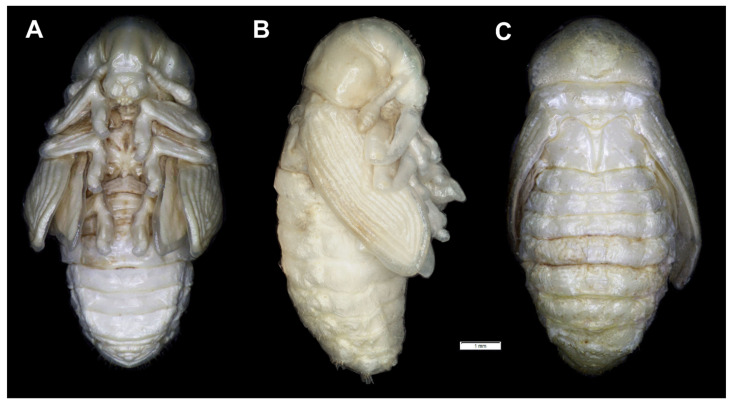
*Rhinocyllus conicus* habitus of pupa. (**A**)—ventral view; (**B**)—lateral view; (**C**)—dorsal view.

**Figure 22 insects-12-00489-f022:**
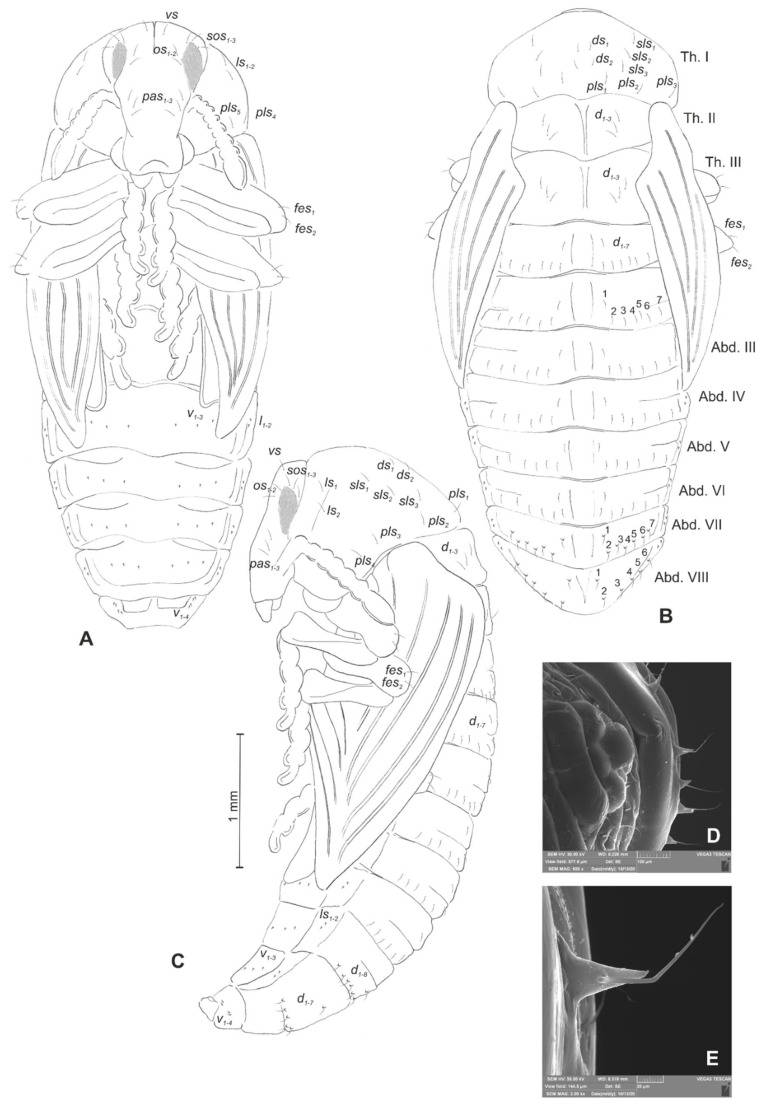
*Rhinocyllus conicus* pupa. (**A**)—ventral view; (**B**)—dorsal view; (**C**)—lateral view; (**D**,**E**)—Abd. VIII, dorsal view, magnification (Th. I–III—pro–, meso– and metathorax, Abd. I–IX—abdominal segments 1–9, ur—urogomphus, setae: *as*—apical, *d*—dorsal, *ds*—discal, *fes*—femoral, *ls*—lateral, *os*—orbital, *pas*—postantennal, *pls*—posterolateral, *rs*—rostral, *sls*—superlateral).

*Chaetotaxy* well-visible, with setae of various lengths from medium to short on abdominal segments VII and VIII placed on small protuberances. Head with three *pas*, two *os* and three *sos*, rostrum without setae. All setae on head equal in length, medium (Figure 22A–C). Pronotum with two *ls*, three *sls*, two *ds*, and four *pls* equal in size. Meso- and metathorax with three medium setae placed medially on dorsum. Each femora with two medium, hair-like setae (Figure 22A). Abdominal segments I–VII with seven (segment VIII with six) medium setae (the first placed medially, the next placed along anterior margins of each of the segments). Setae on abdominal segments VII and VIII placed on protuberances (Figure 22D). Abdominal segment IX with four minute setae placed ventrally. Each lateral part of abdominal segments I–VIII with two minute setae. Ventral parts of abdominal segments I–VIII with three setae. Urogomphia invisible (Figure 22E).

## 4. Discussion

### 4.1. A Comparison with Larvae of Other Lixini

Gosik and Wanat [31] identified the immature stages of the Lixini tribe by the following 16 character sets: “(1) body C–shaped, round in cross-section; (2) head capsule subglobose or oval-shaped; (3) frontal sutures distinct, Y-shaped, extending to a lower pair of stemmata; (4) endocarina line mostly well indicated; (5) head with 5 pairs of *des*, 5 pairs of *fs*, and 2–3 pairs of *les* (except *Lixus strangulatus* Faust, 1883; (6) antennae one-segmented, each located at the end of frontal suture; sensorium conical, more or less elongated; (7) clypeus trapeziform, mostly with 2 pairs of moderately long *cls*; (8) epipharynx usually with 5 pairs of *als*, 3 pairs of *ams*, and 2 pairs of *mes* (except *L. strangulatus*); (9) labral rods well developed, elongated, slightly converging; (10) maxilla with more than 7 *dms*; (11) praelabium heart-shaped or cup-shaped, with a pair of long *plbs*; (12) ligular setae well developed, 2–3 pairs; (13) meso- and metanotum each transversely divided into two lobes (prodorsum with 1 *prs*, postdorsum with 5 *pds*, and 1 *as*); (14) each pedal area well isolated, with more than 5 long *pds*; (15) each of Abd. I–VII with 5–7 *pds*, 1 *dls*, 1 *ss*, 2 *vpls*, and 2 *msts*; (16) Abd. X reduced to four anal lobes of unequal size, dorsal one by far the largest, ventral one very small”. These different traits of mature larvae of Lixini have been confirmed in several descriptions concerning the genus *Larinus* [3,22,32,33,34,35,36,37,38] and the genus *Lixus* [20,21,24,31,33,34,39,40,41] but there were also more exceptions [21,22] than those presented in Gosik and Wanat [31]. All *Larinus* species differ from *Lixus* species, mainly by their (1) U-shaped premental sclerite (vs. *Lixus* species with a trident-shaped premental sclerite), (2) abdominal segments I–VII with three folds (vs. four folds), and (3) pupae with distinct urogomphi (vs. indistinct). However, an exception is represented by most species of the nominotypic subgenus of the genus *Larinus*, whose chaetotaxy on the body is markedly different [22]. Skuhrovec et al. [22] also identified the morphological features of larvae and pupae, which significantly distinguish the subgenus *Larinus* from the other subgenera, *Phyllonomeus* Gistel, 1856 and *Larinomesius*.

### 4.2. Remarks on Lachnaeus

Within the tribe Lixini, the immature stages of *Lachnaeus crinitus* have three unique morphological features: (1) an endocarina is missing (vs. *Larinus* and *Lixus* species with endocarina); (2) the spiracle of thorax looks vestigial, and on high magnification, appears bicameral (vs. the spiracle of the thorax is distinctly bicameral and without the need to control it under high resolution); and (3) the entire dorsal parts of the bodies of the larvae and pupae are distinctly pigmented (vs. the dorsal regions of the larvae and pupae, which not pigmented and sometimes have small dark sclerites). According to the morphological features of immature stages, *Lachaneus crinitus* has a more noticeable affinity with the genus *Larinus* than with the genera close to the genus *Lixus*, as also suggested by adult morphology. *Larinus* species and *Lachnaeus crinitus* share two unique morphological features: (1) a U-shaped premental sclerite (vs. *Lixus* species, which have a trident-shaped premental sclerite similar to *Rhinocyllus* species and some Cleonini species); and (2) abdominal segments I–VII with three folds (vs. four folds).

### 4.3. Remarks on Rhinocyllus

The immature stages of *R. conicus* and *R. alpinus* have several specific features, the combination of which is completely unique within the Lixini tribe: (1) a trident-shaped premental sclerite (identical to *Lixus* species, but other genera have a U-shaped premental sclerite); (2) abdominal segments on larva with five or six *pds* (vs. more *pds* on abdominal segments); (3) meso- and metanotum on pupa with only three *d* (vs. more than five *d*); and finally, (4) pupa “invisible” urogomphi (vs. distinct or sometimes very small indistinct). Based on our data on *Rhinocyllus*, the type genus of the tribe Rhinocyllini, although without information on *Bangasternus* and *Microlarinus*, we can conclude that this tribe is not supported by unique character states (apomorphies) in the immature stages. The pupae of both species of *Rhinocyllus* seem to show a closer relationship with the genus *Lixus* than to *Larinus,* in contrast to what is suggested by the characters of the adults, because *Rhinocyllus* species do have “invisible” urogomphi, while *Lixus* species do not have urogomphi, or they are very small and almost indistinct, unlike the pupae of *Larinus* and *Lachnaeus*, which always have quite noticeable urogomphi. Yet, *Rhinocyllus* is similar to *Larinus obtusus* and *Lachnaeus*, having only five or six *pds* on abdominal segments instead of more *pds* on abdominal segments, similar to other “large” species of *Larinus* and *Lixus*. In this case, however, it is possible that this character might only be related to the shrinking of the body and may not be phylogenetically important, as we have observed in other unrelated tribes, such as the Tychiini of the subfamily Curculioninae [42].

The primary differences between the larvae and pupae of *Rhinocyllus alpinus* and *R. conicus* currently are those that distinguish the two subgenera *Rhinocyllus* and *Rhinolarinus* and are as follows: (1) *des_5_* very short (vs. *R. conicus* with medium *des_5_*), (2) *des_5_* close to *des_3_* and *des_4_* (vs. *des_5_* separate), (3) *lrs_2_* as long as *lrs_1_* (vs. *lrs_2_* twice as long as *lrs_1_*), (4) head of the pupa with one *sos* (vs. with three *sos* and one vs), (5) rostrum of the pupa with two *rs* (vs. without *rs*), and (6) dorsum of the pupal abdomen with eight minute *d* (vs. with seven short *d*).

## 5. Conclusions

The complex of these new data have allowed us to support that (1) *Lachnaeus* and *Rhinocyllus* are two valid genera that are different from *Larinus*, (2) Rhinocyllini is not a tribe different from Lixini, and (3) the separation of *Rhinocyllus* into two subgenera is rational. Therefore, it seems obvious why it will be important to increase the exiguous number of species of this tribe with known immature stages in order to address other unresolved questions, as already shown in other tribes or subfamilies (see Otiorhynchinini [43], Cionini [44], Mecinini [45,46], Smicronychini [47], Tychiini [42] and Cossoninae [48]). It is clearly confirmed that a detailed description of immature stages is a very important component for a better understanding of generic taxonomy but also tribal taxonomic and phylogenetic relationships within Lixinae, as well as within other subfamilies (see above). It is not to be neglected that in the Lixini, these new data may have particular relevance, because *Rhinocyllus conicus* and several other species of the genera *Lixus* and *Larinus* have a practical or at least potential use as biological control agents against invasive and noxious weeds (e.g., *Carduus*, *Cirsium*, *Tanacetum*).

## Figures and Tables

**Figure 3 insects-12-00489-f003:**
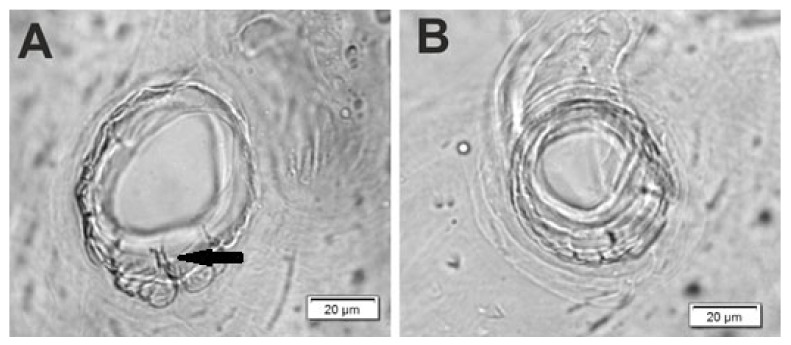
*Lachnaeus crinitus* mature larva, spiracles. (**A**)—spiracle of prothorax; (**B**)—spiracle of abdominal segment I.

**Figure 4 insects-12-00489-f004:**
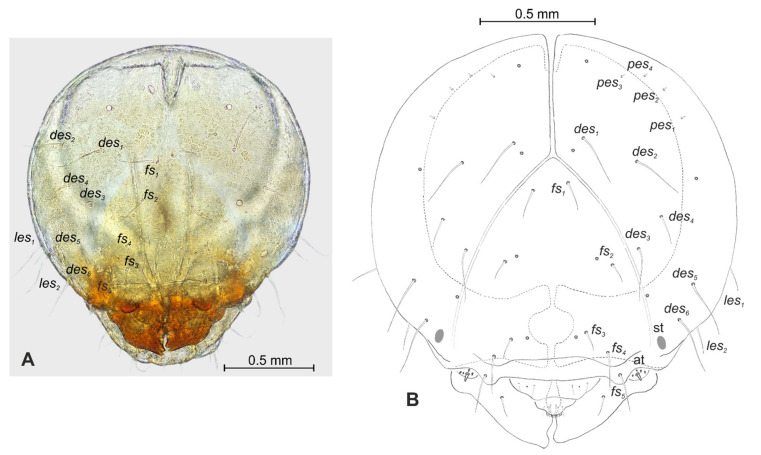
*Lachnaeus crinitus* mature larva, head. (**A**)—frontal view of head, photo; (**B**)—frontal view of head, scheme; (at—antenna, st—stemmata, setae: *des*—dorsal epicranial, *fs*—frontal, *les*—lateral epicranial, *pes*—postepicranial).

**Figure 5 insects-12-00489-f005:**
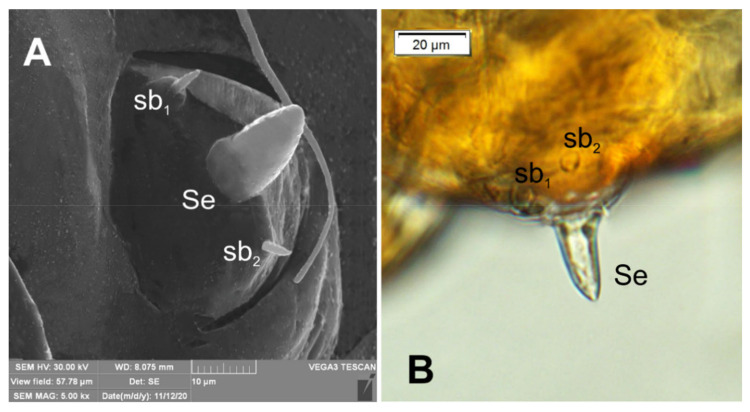
*Lachnaeus crinitus* mature larva, antenna. (**A**)—antenna SEM photo; (**B**)—antenna photo; (sa—sensillum ampullaceum, sb—sensillum basiconicum, Se—sensorium).

**Figure 6 insects-12-00489-f006:**
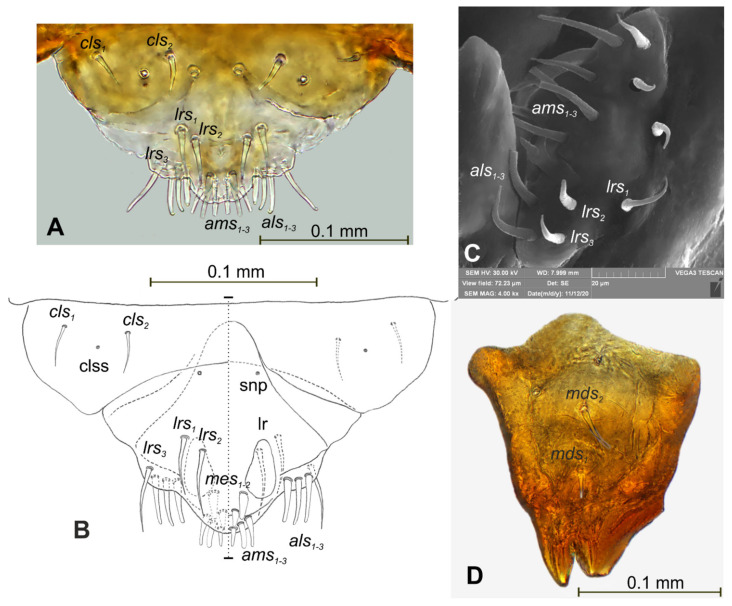
*Lachnaeus crinitus* mature larva, mouthparts. (**A**)—clypeus, labrum, and epipharynx, photo; (**B**)—clypeus, labrum and epipharynx, scheme; (**C**)—clypeus, labrum, and epipharynx, SEM photo; (**D**)—right mandible (clss—clypeal sensorium, lr—labral rods, snp—sensillae pore, setae: *als*—anterolateral, *ams*—anteromedial, *cls*—clypeal, *lrs*—labral, *mds*—*mandibular*, *mes*—median).

**Figure 7 insects-12-00489-f007:**
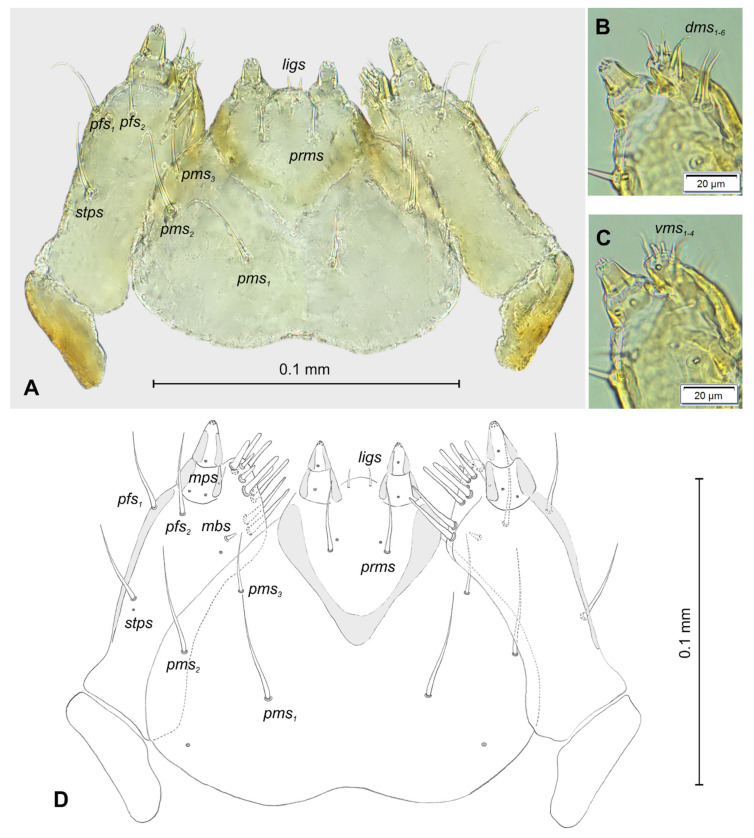
*Lachnaeus crinitus* mature larva, maxillolabial complex. (**A**)—maxillolabial complex, ventral aspect, photo; (**B**)—maxillolabial complex, ventral aspect, scheme; (**C**)—apical part of right maxilla, dorsal aspect; (**D**)—apical part of right maxilla, ventral aspect (setae: *dms*—dorsal malar, *ligs*—ligular, *mbs*—malar basiventral, *mps*—maxillary palp, *pfs*—palpiferal, *prms*—prelabial, *pms*—postlabial, *stps*—stipal, *vms*—ventral malar).

**Figure 8 insects-12-00489-f008:**
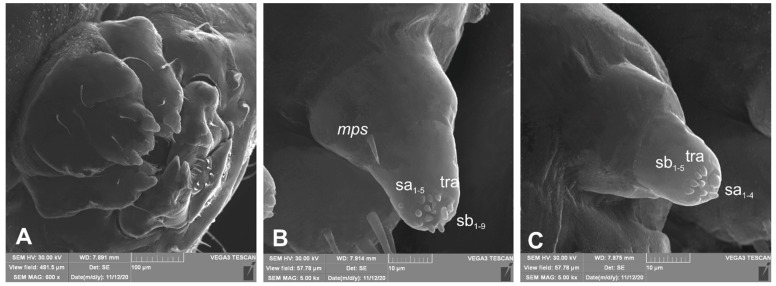
*Lachnaeus crinitus* mature larva, maxillolabial complex, SEM photo. (**A**)—maxillolabial complex, ventral aspect; (**B**)—apical part of maxillary palp; (**C**)—apical part of labial palp (ds—digitiform sensillum, sa—sensillum ampullaceum, sb—sensillum basiconicum, tra—terminal receptive area).

**Figure 9 insects-12-00489-f009:**
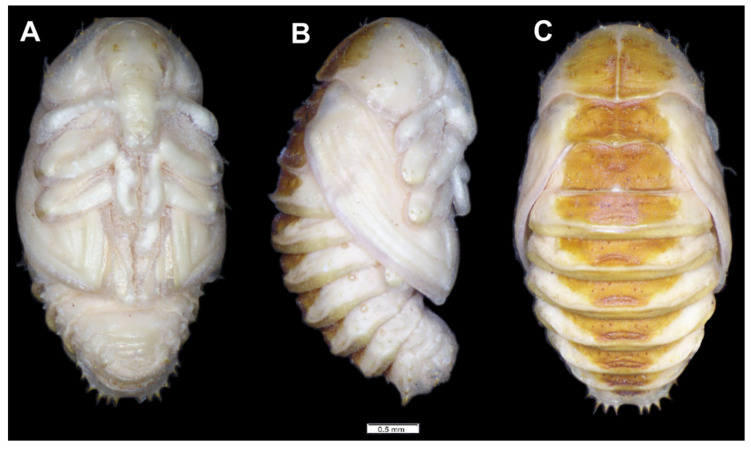
*Lachnaeus crinitus* habitus of pupa. (**A**)—ventral view; (**B**)—lateral view; (**C**)—dorsal view.

**Figure 10 insects-12-00489-f010:**
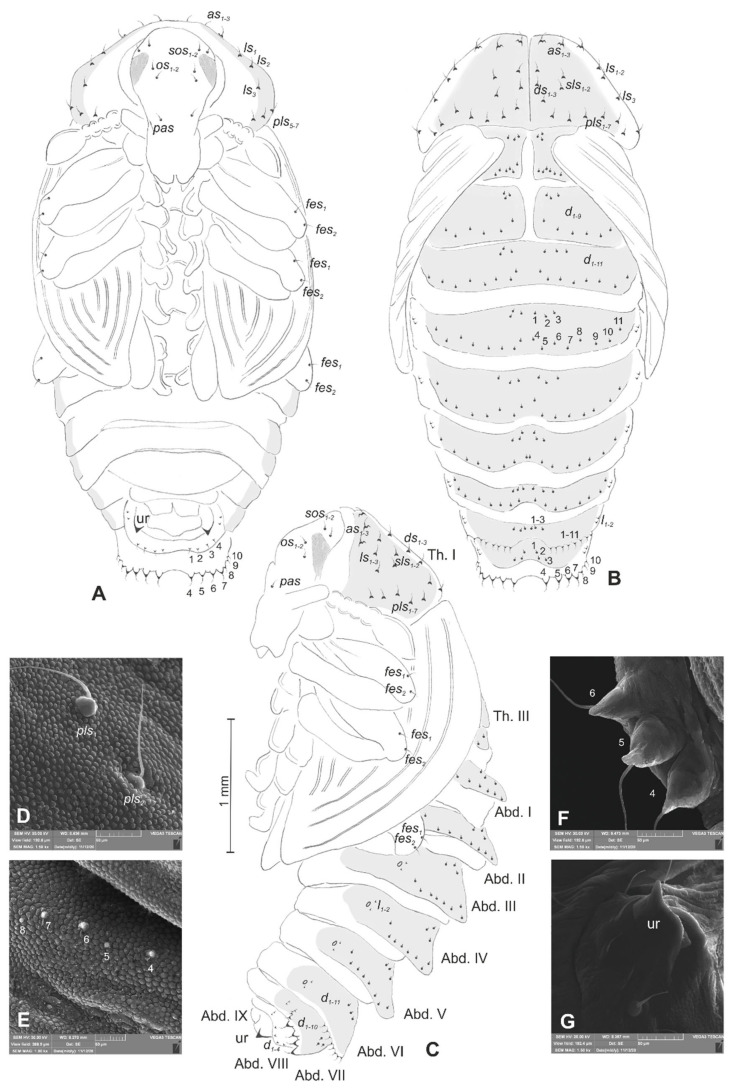
*Lachnaeus crinitus* pupa. (**A**)—ventral view; (**B**)—dorsal view; (**C**)—lateral view; (**D**)—pronotum, dorsal view, magnification; (**E**)—Abd. I, dorsal view, magnification; (**F**)—Abd. VII, dorsal view, magnification; (**G**)—urogomphus, magnification (Th. I–III—pro-, meso- and metathorax, Abd. I–IX—abdominal segments 1–9, ur—urogomphus, setae: *as*—apical, *d*—dorsal, *ds*—discal, *fes*—femoral, *ls*—lateral, *os*—orbital, *pas*—postantennal, *pls*—posterolateral, *rs*—rostral, *sls*—superlateral).

**Figure 11 insects-12-00489-f011:**
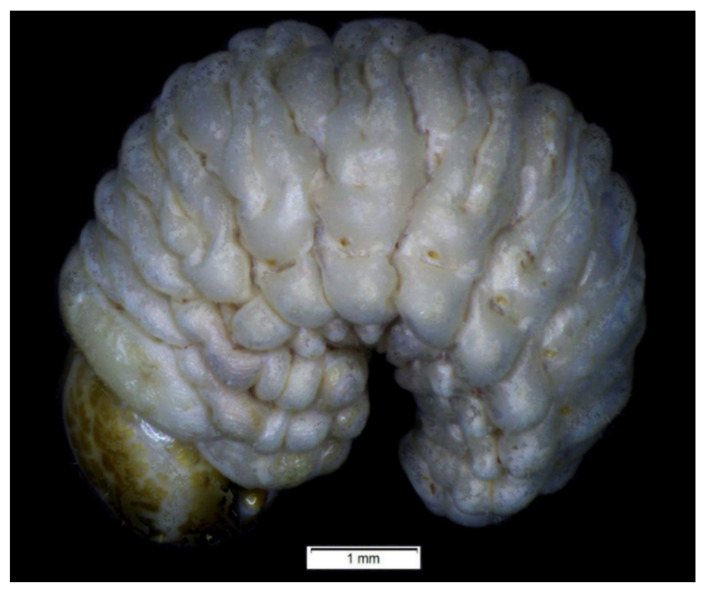
*Rhinocyllus alpinus* habitus of mature larva, lateral view.

**Figure 13 insects-12-00489-f013:**
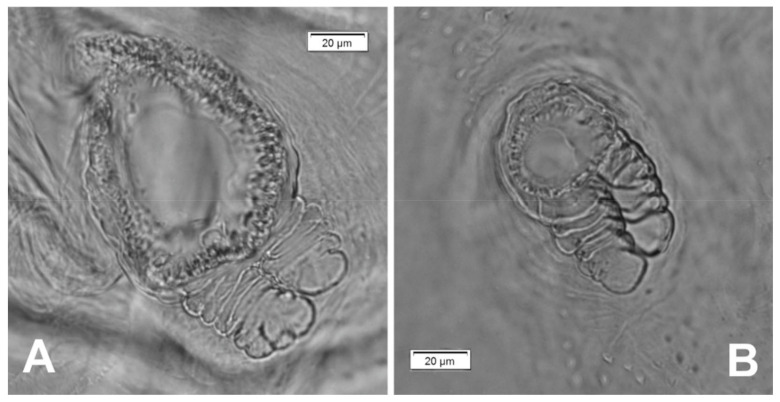
*Rhinocyllus alpinus* mature larva, spiracles. (**A**)—spiracle of prothorax; (**B**)—spiracle of abdominal segment I.

**Figure 14 insects-12-00489-f014:**
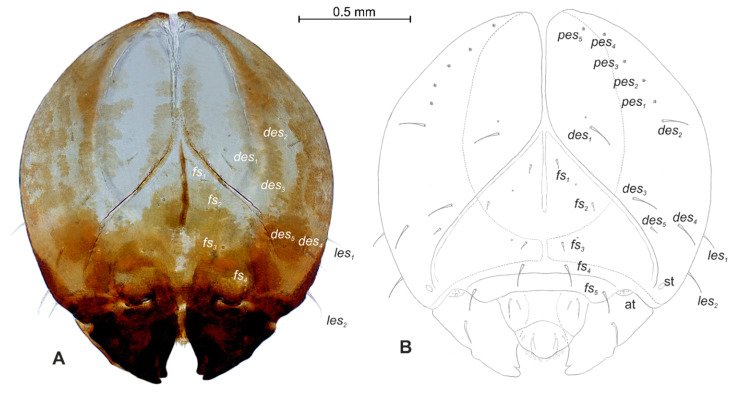
*Rhinocyllus alpinus* mature larva, head. (**A**)—frontal view of head, photo; (**B**)—frontal view of head, scheme; (at—antenna, st—stemmata, setae: *des*—dorsal epicranial, *fs*—frontal, *les*—lateral epicranial, *pes*—postepicranial).

**Figure 15 insects-12-00489-f015:**
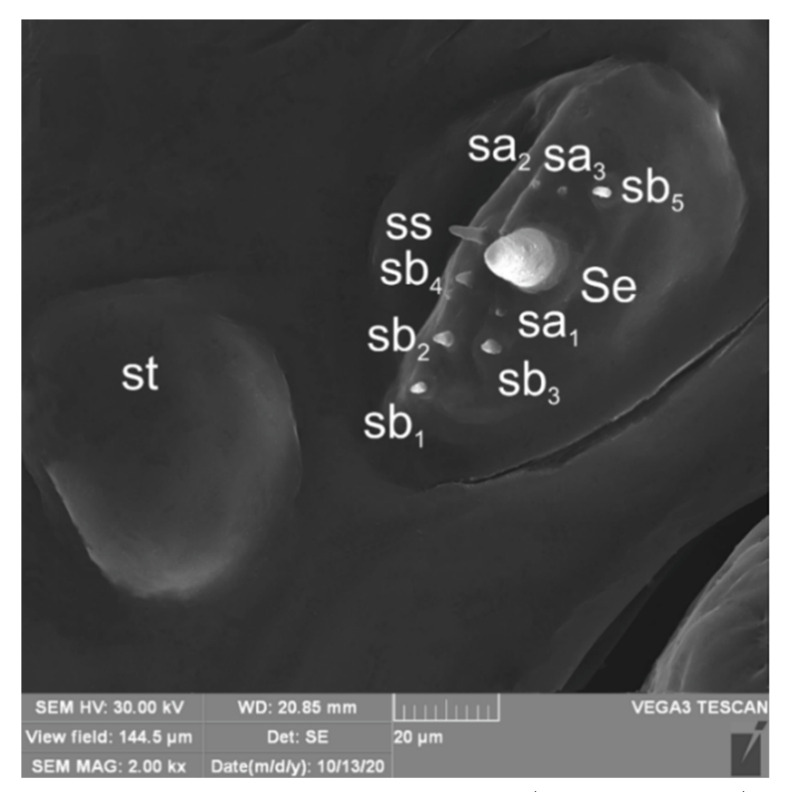
*Rhinocyllus alpinus* mature larva, antenna SEM photo (sa—sensillum ampullaceum, sb—sensillum basiconicum, Se—sensorium, st—stemmata).

**Figure 16 insects-12-00489-f016:**
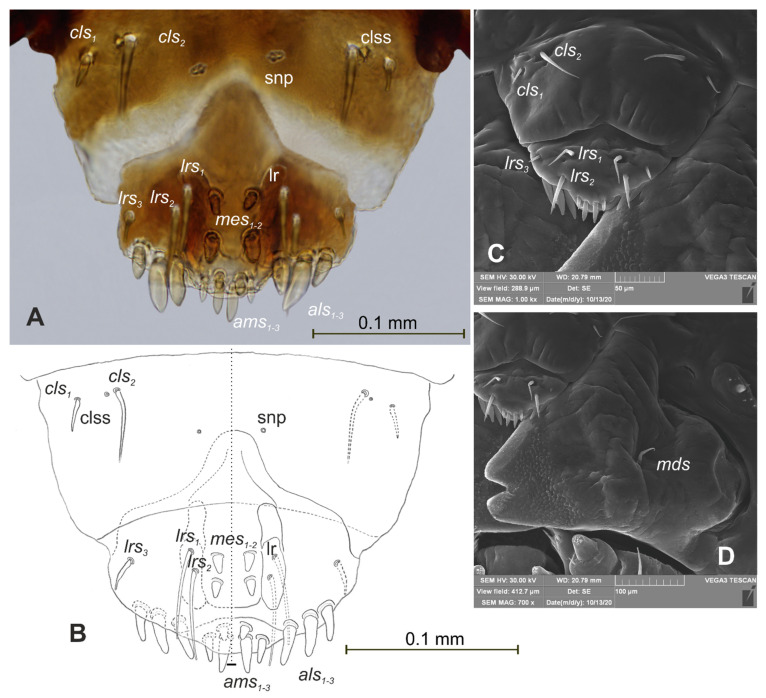
*Rhinocyllus alpinus* mature larva, mouthparts. (**A**)—clypeus, labrum, and epipharynx, photo; (**B**)—clypeus, labrum and epipharynx, scheme; (**C**)—clypeus, labrum, and epipharynx, Scheme 17. *Rhinocyllus alpinus* mature larva, maxillolabial complex. (**A**)—maxillolabial complex, ventral aspect, photo; (**B**)—maxillolabial complex, ventral aspect, scheme; (**C**)—apical part of right maxilla, dorsal aspect; (**D**)—apical part of right maxilla, ventral aspect; (setae: *dms*—dorsal malar, *ligs*—ligular, *mbs*—malar basiventral, *mps*—maxillary palp, *pfs*—palpiferal, *prms*—prelabial, *pms*—postlabial, *stps*—stipal, *vms*—ventral malar).

**Figure 17 insects-12-00489-f017:**
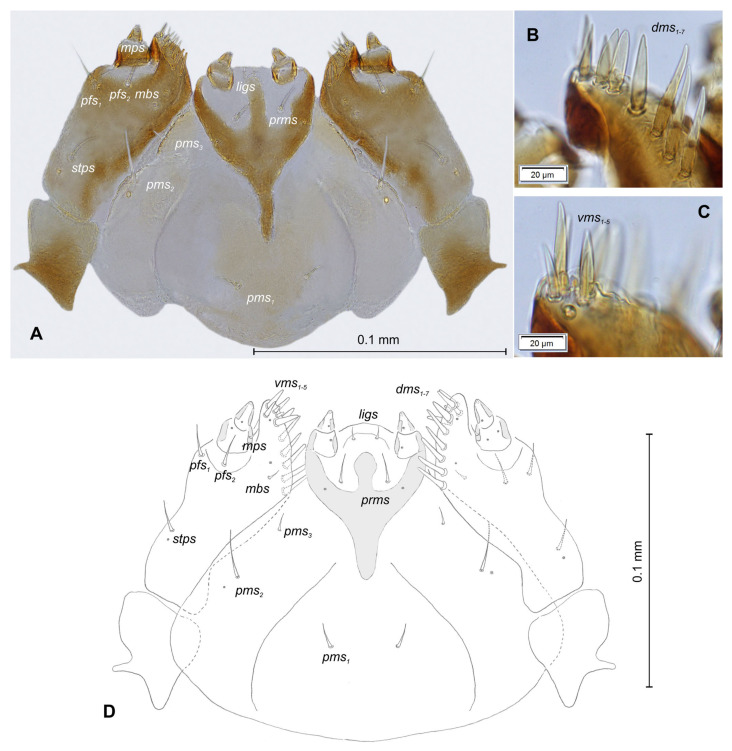
*Rhinocyllus alpinus* mature larva, maxillolabial complex. (**A**)—maxillolabial complex, ventral aspect, photo; (**B**)—maxillolabial complex, ventral aspect, scheme; (**C**)—apical part of right maxilla, dorsal aspect; (**D**)—apical part of right maxilla, ventral aspect; (setae: *dms*—dorsal malar, *ligs*—ligular, *mbs*—malar basiventral, *mps*—maxillary palp, *pfs*—palpiferal, *prms*—prelabial, *pms*—postlabial, *stps*—stipal, *vms*—ventral malar).

**Figure 18 insects-12-00489-f018:**
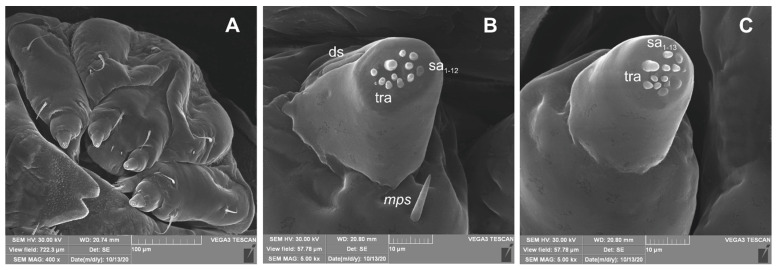
*Rhinocyllus alpinus* mature larva, maxillolabial complex, SEM photo. (**A**)—maxillolabial complex, ventral aspect; (**B**)—apical part of maxillary palp; (**C**)—apical part of labial palp (ds—digitiform sensillum, sa—sensillum ampullaceum, sb—sensillum basiconicum, tra—terminal receptive area).

**Table 1 insects-12-00489-t001:** Results of measurements of larvae and pupae of *Lachnaeus* and *Rhinocyllus***.** All data in (mm), ^n^: number of exemplars, mv = medium value.

Stages	*Lachnaeus crinitus*			*Rhinocyllus alpinus*			*Rhinocyllus conicus*		
**Larva**	**HW**	**BL**	**BW**	**HW**	**BL**	**BW**			
	0.90 ^1^, 1.25 ^1^, 1.40 ^1^, 1.50 ^1^, 1.75 ^1^	2.50 ^1^, 3.25 ^1^, 4.00 ^2^, 4.25 ^1^	1.00 ^1^, 1.20 ^4^	1.20 ^3^, 1.25 ^19^, 1.30 ^7^ (mv: 1.25)	3.25 ^2^, 3.50 ^5^, 4.00 ^2^, 4.25 ^5^, 4.50 ^10^, 4.80 ^2^, 5.00 ^2^ (mv: 4.50)	1.75 ^4^, 2.00 ^6^, 2.25 ^5^, 2.50 ^8^, 2.75 ^5^, 3.00 ^1^ (mv: 2.80)			
**Pupa ♀**	**THW**	**BL**	**BW**	**THW**	**BL**	**BW**	**THW**	**BL**	**BW**
	1.55 ^1^	6.50 ^1^	3.50^1^	2.10 ^2^, 2.30 ^2^ (mv: 2.20)	5.50 ^1^, 6.00 ^3^ (mv: 6.00)	3.30 ^1^, 3.40 ^3^ (mv: 3.40)	1.60 ^2^, 1.80 ^2^, 2.00 ^1^ (mv: 1.80)	4.50 ^1^, 5.20 ^3^, 5.50 ^1^ (mv: 5.20)	2.20 ^2^, 2.40 ^2^, 2.90 ^1^ (mv: 2.40)
**Pupa ♂**	1.50 ^1^, 1.70 ^4^ (mv: 1.70)	5.00 ^1^, 6.40 ^4^ (mv: 6.00)	3.00 ^1^, 3.50 ^4^ (mv: 3.40)	2.10 ^10^, 2.20 ^2^ (mv: 2.10)	5.00 ^3^, 5.50 ^4^, 6.50 ^5^ (mv: 6.50)	3.00 ^7^, 3.40 ^3^, 3.50 ^2^ (mv: 3.00)	1.50 ^3^, 2.00 ^6^, 2.10 ^1^ (mv: 2.00)	4.50 ^1^, 5.20 ^3^, 6.10 ^4^, 7.00 ^2^ (mv: 6.10)	2.10 ^4^, 3.00 ^6^, 3.40 ^2^ (mv: 3.00)

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
