# Peer review of "On the Affinities and Systematic Position of Lachnaeus Schoenherr and Rhinocyllus Germar in the Tribe Lixini (Coleoptera: Curculionidae: Lixinae) Based on the Morphological Characters of the Immature Stages"

_insects, 2021, doi:10.3390/insects12060489_

Round 1

Reviewer 1 Report

Very nice manuscript, that deserves to be published. I only detected that in the figure captions "Lachnaeus" is incorrectly spelled "Lachaenus". 

The ms addresses a relevant and interesting question, although only for specialists in this beetle taxon. It is well written, and I find the text quite clear. In general, it represents a fine contribution to the systematics of the subfamily Lixinae of Curculionidae.

Author Response

Thanks for the very positive review, and for the little comments that will help improve our manuscript.

Reviewer 2 Report

This is a very interesting, nicely illustrated and very carefully prepared manuscript which brings critical new data about the Lixinae weevils. These are not only applicable for the systematics, in the line as the authors comment directly in the text; given the phytosanitary importance of the lixine beetles, it is also very good that the paper presents the morphological characters of larvae and pupae and will allow for their identification. I only have few minor comments in the attached PDF, the only "serious" problems seems to me the formatting of the table, which is at the moment very confusing. Otherwise the manuscript is ready to be published.

Author Response

Thank you for a very positive and constructive review.

All comments were accepted and resolved directly in the text.

Table - has been changed and its structure has been simplified.

Yes, the material of both larvae and pupae was collected directly in the field, as mentioned in the first paragraph of the Material and Methods.

Not only larvae but also pupae and adults were always collected on the host plant. Pupae already have the characteristics of adults and can be easily identified in a group, so it was not a problem to identify these specimens. 

Reviewer 3 Report

I found this manuscript to be very well written and the figures were carefully created. 

Author Response

(The authors gave the same response as above.)
